# A Reverse Osmosis Process to Recover and Recycle Trivalent Chromium from Electroplating Wastewater

**DOI:** 10.3390/membranes12090853

**Published:** 2022-08-31

**Authors:** Roxanne Engstler, Jan Reipert, Somayeh Karimi, Josipa Lisičar Vukušić, Felix Heinzler, Philip Davies, Mathias Ulbricht, Stéphan Barbe

**Affiliations:** 1Faculty of Applied Natural Sciences, TH Köln—University of Applied Sciences, 51379 Leverkusen, Germany; 2Department of Technical Chemistry II, University of Duisburg-Essen, 45141 Essen, Germany; 3BIA Kunststoff-und Galvanotechnik GmbH & Co. KG, 42655 Solingen, Germany; 4School of Engineering, University of Birmingham, Birmingham B15 2FG, UK

**Keywords:** electrolyte recovery, reverse osmosis, closed-loop, wastewater treatment, electroplating

## Abstract

Electroplating generates high volumes of rinse water that is contaminated with heavy metals. This study presents an approach for direct metal recovery and recycling from simulated rinse water, made up of an electroplating electrolyte used in industry, using reverse osmosis (RO). To simulate the real industrial application, the process was examined at various permeate fluxes, ranging from 3.75 to 30 L·m^−2^·h^−1^ and hydraulic pressures up to 80 bar. Although permeance decreased significantly with increasing water recovery, rejections of up to 93.8% for boric acid, >99.9% for chromium and 99.6% for sulfate were observed. The final RO retentate contained 8.40 g/L chromium and was directly used in Hull cell electroplating tests. It was possible to deposit cold-hued chromium layers under a wide range of relevant current densities, demonstrating the reusability of the concentrate of the rinsing water obtained by RO.

## 1. Introduction

Electroplating chromium on thermoplastics requires a multilayer electrodeposition treatment with several interim rinsing steps. In the process, these rinses become highly contaminated with heavy metals as well as all other components from the plating electrolyte. In this way, several thousand liters of wastewater are produced per week in a typical plating facility.

Due to the recent ban of carcinogenic Cr(VI) in the European REACH decree [1], industrial applications transition to far less toxic trivalent Cr(III) [2]. Establishing new plating processes also brings the need to evaluate costs for connected processes, such as wastewater disposal. The utilization of water and heavy metals from the rinse water would not only reduce costs for disposal and make-up of plating baths but also fit the claim of the European Green Deal for a circular economy [3].

Regarding chromium electroplating, the separation of metal ions and water has already been the focus of research. Most commonly, chromium is removed via chemical precipitation forming hydroxides [4,5,6], adsorbed on activated carbon [7,8], adsorbed on ion exchange resins [9,10] or separated in membrane processes [11] via ultrafiltration (UF) [12], nanofiltration (NF) [13,14] or reverse osmosis (RO) [15,16,17,18,19]. The last one has been recognized as a potential technology for the purification of electroplating wastewater for a long time [11,13,15,16,17,18,19,20,21,22,23,24,25,26,27,28,29,30], but in all these publications, only water, if at all, is recycled in the overall process. The appropriate recovery and recycling of electrolytes from wastewater have not yet been extensively investigated [24,31].

Typical chromium(III) electrolytes for decorative purposes contain a chromium source such as $CrCl_{3}\cdot 6H_{2}O$, $Cr_{2}{\left(SO_{4}\right)}_{3}$ or $Cr\left(SO_{4}\right)\left(OH\right)$ with organic acids and their salts (oxalic acid, malic acid [32], glycolic acid, malonic acid [13], acetic acid [14] or formic acid [15]) as complexing ligands and large quantities of boric acid ($H_{3}BO_{3}$) as buffer substance [14,16]. Sulfuric acid and potassium hydroxide are used for pH adjustments since most electrolytes are used under acidic conditions [17]. Sodium sulfate or aluminum sulfate act as conducting salts. Sodium saccharin, allyl sulfonic salt or fluoride reduce the co-deposition of carbon for a bright finish of the deposited layer [15]. Surfactants reduce the surface tension of the bath to minimize losses due to drag-out [2]. The washing of freshly plated parts in between plating steps contaminates large quantities of rinse water with these substances.

The removal of boric acid from aqueous solutions is a known challenge [33,34,35]. Boric acid is a weak Lewis acid, and below pH 9.23, boric acid exists mainly in its undissociated, neutral form, with a small hydrated radius. This is the main reason for low boric acid rejection in RO processes [35]. For example, a rejection of 85–90% was found for the FILMTEC SW30-380 membrane in a single pass at pH 8. Due to the risk of chromium hydroxide precipitation, pH increase is no option for the rinse water treatment.

In chromium electroplating baths, boric acid is used in concentrations of 0.5–1.2 mol/L (31–74 g/L) [36,37,38]. To increase solubility (55 g/L at 25 °C [33]), these electrolytes are used at temperatures of 25–55 °C [39,40]. The present sulfate further increases boric acid solubility [41]. Rinse water tanks in electroplating are not heated, and higher diffusion rates at elevated temperatures would lead to lower rejection by RO membranes [42]. Overall, limited solubility and permeation of boric acid through the membrane were identified as challenges in the successful recycling of electroplating effluents.

Organic substances, such as malic acid and saccharin, are likely to adhere to a RO membrane surface; the accumulation of these substances can lead to fouling, successively decreasing membrane permeance [43]. Organic substances cause up to 30% of the cost for the make-up of an electroplating electrolyte. The high concentrations of malic acid and saccharin are essential for its functionality. Therefore, the possibility for direct recovery, as considered for other electrolytes [24,31,44,45], has to be determined.

Although direct recovery of electroplating electrolyte components has been reported, the proof of reusability of such RO concentrates with a Hull cell test was not previously published. The close observation of the influence of RO process parameters on membrane performance, as well as the in-depth analysis of the obtained concentrates and permeates, has to be performed to obtain a detailed picture of the process.

Large amounts of organic additives, able to cause membrane fouling, are mostly referred to as the main limiting factor for direct, membrane-based recovery treatments [46,47]. Nevertheless, such treatments have been researched, and some have found application in the industry [24].

Castelblanque presented the results of an RO-based plant operating in an Italian electroplating facility, which was used to treat nickel rinse water containing 20 g/L Ni and 15 g/L boric acids. Although it was stated that the wastewater contained organic brighteners in variable portions, the successful recovery and reuse of the rinse water were documented. The use of RO instead of nanofiltration was motivated by the significantly better rejection of boric acid [24]. The need for high metal and boric acid contents matches the process of this work.

Schoeman reported on the treatment of chromium electroplating effluent using RO at 40 bar, creating a concentrate containing a total of 24.4 g/L Cr, achieving a concentration factor of 13.3, and a permeate with 25.2 mg/L Cr [44]. UF was used as a pretreatment of the effluent. Although stated as originating from an electroplating process, neither the buffer boric acid nor organic compounds were monitored. A chromium rejection of 98.6% was achieved, and the author reported that no fouling was observed. The reuse of concentrate and permeate in the electroplating process was suggested [44].

In this work, the feasibility of processing trivalent chromium electroplating wastewater in a straightforward RO treatment and to recycle the generated retentate into the electroplating process was investigated. Therefore, a commercially available diluted Cr(III) electrolyte was used as simulated rinse water and feed solution in a reverse osmosis lab unit (ROLU). The concentration of this feed was step-wise increased in a series of consecutive experiments using economically relevant pressures (≤80 bar) and fluxes (≤30 LMH). The concentrations of chromium, boric acid and sulfate in the generated concentrates and permeates were analyzed with inductively coupled plasma atomic emission spectroscopy (ICP-OES), and their surface tension was determined. The membrane rejection values were calculated with the results of ICP-OES analysis for different permeate fluxes and feed concentrations. Hydraulic pressures and osmotic pressures of the RO concentrates, as well as permeate fluxes, were used to calculate the driving force and membrane permeance, respectively. The usability of the final RO concentrate for electroplating was investigated with Hull cell tests. Possible causes for the observed decreasing membrane permeance were discussed.

## 2. Materials and Methods

### 2.1. Reverse Osmosis in a High-Pressure Membrane Plant

The commercial mixtures and solutions (Make-Up, Buffer, Replenisher, Wetting Agent) to prepare an industrial Cr(III) electrolyte were kindly provided by BIA Kunststoff- und Galvanotechnik GmbH & Co. KG. One liter of electrolyte was prepared in a 2L-Erlenmeyer flask by adding 400 mL of deionized water and 300 mL of Make-Up solution (containing chromium hydroxide sulfate) and heated to 65 °C. A total of 190 g Buffer (containing potassium sulfate and boric acid) was added slowly under stirring until dissolved completely. Then, 20 mL of Replenisher (containing saccharin) was added, and the pH value of the solution was adjusted to 4.7 with potassium hydroxide (1 M) and sulfuric acid (70%) (both p.a.; Merck KGaA, Darmstadt, Germany). Then, 1 mL of Wetting Agent (containing sodium 1,4-bis(1,3-dimethylbutyl)sulphonatosuccinate was added, and the solution was diluted (1:10) with deionized water at 45 °C. After cooling to room temperature, the solution was filtrated through a 0.2 µm cellulose acetate filter (Sartorius Stedim Biotech GmbH, Goettingen, Germany) and used as the feed solution. The concentration of the most important components in the electrolyte and feed solution are listed in Table 1.

All RO trials were performed at 25 °C, with a maximum cross-flow velocity of 0.25 m/s and a maximum hydraulic pressure of 80 bar. More information on the membrane specifications is listed in Table 2.

The ROLU used in this work is depicted in Figure 1. The feed solution is pumped (P-001) from the mixing tank (D-001; 1.1 L) through two flat-sheet membrane cells (ME-001 and ME-002) connected in series, operating in cross-flow. The retentate stream is led back to the mixing tank (D-001) where the pH value, electrical conductivity, temperature and liquid level are monitored. The described ROLU is mainly controlled by means of a flux-driven operation mode. For this purpose, the permeate flow is defined by a set point within the software. The ROLU may be operated in two operation modes:A steady-state mode, where the resulting permeate and retentate are led back into the mixing tank. The two streams are mixed to restitute the original feed stream, which is then recirculated. In this steady-state mode, the feed concentration is constant.A concentration mode (dashed line in Figure 1) where the permeate is discharged from the system and the retentate is led back to the mixing tank leading to an increase in the concentration of soluble compounds. Volume loss is compensated by feed solution make up.

In this research, a series of steady-state RO experiments were conducted at different permeate fluxes (30, 25, 20, 15 and 10 $L\cdot m^{-2}\cdot h^{-1}$). For each permeate flux, steady-state conditions were maintained for 60–90 min and permeate samples (P) were taken (e.g., P1.1–P1.4). Concentrate samples (K) were taken at the beginning and the end of every steady-state experiment (e.g., K1.1 and K1.2). The electrical conductivity of concentrate and permeate streams, as well as the pH of the concentrate, were continuously monitored.

After each series of RO experiments, the feed concentration was increased in concentration mode until the hydraulic pressure increased by 10 bar at maximum flux. Subsequently, a new series of steady-state RO experiments at different fluxes were performed, as previously stated. After the maximum hydraulic pressure (80 bar) was reached, further concentration increase was performed at reduced permeate flux. A steady-state mode with reduced flux was also used overnight to avoid static permeate-side backpressure. No membrane cleaning was performed in between the various experiments.

The degree of concentration increase in the RO retentate is expressed as the concentration factor of chromium (cf):(1)$$cf=\frac{\beta {\left(Cr\right)}_{max}}{\;\beta {\left(Cr\right)}_{i}}$$
where $\beta {\left(Cr\right)}_{max}$ is the mass concentration of chromium in the final concentrate (8.4 g/L) and $\beta {\left(Cr\right)}_{i}$ is the mass concentration of chromium of a previous concentrate sample. Overall, 14 RO experiments were conducted, starting with the simulated rinse water (feed, 0.77 g/L Cr(III), cf 1.00) and step-wise increasing the retentate concentration. The procedure was repeated until the maximum possible flux was $7.5\;L\cdot m^{-2}\cdot h^{-1}$ (80 bar; cf 10.92; see Results and Discussion).

### 2.2. Permeate Flux, Osmotic Pressure and Membrane Permeance

The volume flow rate of permeate ${\dot{V}}_{P}$ was monitored with a digital inline flowmeter using thermal flow measurement (LIQUI-FLOW™, BRONKHORST DEUTSCHLAND NORD GmBH, Kamen, Germany). The total membrane area $A_{M}$ the permeate flux $J_{P}$ was calculated:(2)$$J_{P}=\frac{{\dot{V}}_{P}}{\;A_{M}}$$

The osmotic pressure of the RO concentrates (ΔΠ) was calculated from water activity ($a_{w}$) analyses obtained with the vapor pressure osmometer AQUALAB^®^ 4TEV (METER Group, Inc., Pullman, WA, USA). The osmometer is equipped with a chilled-mirror dew point sensor with an accuracy of ±0.003 $a_{w}$. The non-simplified van‘t Hoff equation (Equation (3)) was used [48]:(3)$$\Delta \Pi =-\left(\frac{RT}{V_{m}}\right)\cdot \ln a_{w}$$
where R is the universal gas constant, T is the temperature and $V_{m}$ is the partial molar volume of water.

Ideally $J_{P}$ is equal to the permeation flux of water, depending on the permeance A and the difference in hydraulic pressure Δp and transmembrane osmotic pressure difference Δπ [49]:(4)$$J_{P}=A\left(\Delta p-\Delta \pi \right)$$

The driving force was calculated using Δ*p* − Δ*π*. The osmotic pressures of the permeates were neglected due to their comparably small contribution. Permeate flux and driving force were then used to calculate membrane permeance:(5)$$A=\frac{J_{P}}{\left(\Delta p-\Delta \pi \right)}$$

### 2.3. Salt Rejection and Permeance Tests

Salt rejection and permeance of the membranes were investigated before and after the RO experiments with 2% NaCl solution (Π=19.1 bar). The electrical conductivity of concentrate $\sigma_{NaCl\_C}$ and permeate $\sigma_{NaCl\_P}$ were measured in steady-state (after 2 h) with conductivity meters (ACS-Z with K = 0.1 and K = 1.0, both KOBOLD Messring GmbH, Hofheim, Germany) to calculate the salt rejection $R_{NaCl}$ using Equation (6)):(6)$$R_{NaCl}=\frac{\sigma_{NaCl\_C}-\sigma_{NaCl\_P}}{\sigma_{NaCl\_C}}\cdot 100$$

Due to the significant change in permeance after the RO experiments, two different parameters were checked: First, the same hydraulic pressure was applied as prior to the RO experiments; in a second test, the same permeate flux was applied. For both cases, permeance and salt rejection were determined.

### 2.4. Inductively Coupled Plasma Optical Emission Spectrometry

Spectroscopic determination of total chromium, sulfur and boron was performed with inductively coupled plasma optical emission spectrometry (ICP-OES, SPECTROGREEN, SPECTRO Analytical Instruments GmbH, Kleve, Germany) for RO permeate and concentrate samples. The contents of sulfate and boric acid were derived from these data. Permeates were measured directly, whereas concentrates were determined in dilution (1:100). Each measurement was performed three times, and average values were used to estimate concentrations.

The percentage membrane rejection $R_{i}$ for a component i was calculated with:(7)$$R_{i}=\frac{\beta_{iC}-\beta_{iP}}{\beta_{iC}}\cdot 100$$
where $\beta_{iC}$ is the mass concentration of the component in the concentrate and $\beta_{iP}$ is the mass concentration of the same component in the permeate.

### 2.5. Surface Tension

The surface tension of RO concentrates and permeates was determined with the bubble pressure tensiometer SITA DynoTester+ (SITA Messtechnik GmbH, Dresden, Germany). Measurements were performed at room temperature (22.6–23.6 °C) with a bubble lifetime of 4000 ms. Each analysis was performed three times, and average values are reported.

### 2.6. Hull Cell Testing

The plating ability of the final RO concentrate on the dependency of the current density was studied using a 250 mL polymethylmethacrylate temperature-controlled Hull cell, as described in the German standard DIN 50957-1 (see Figure 2) [50].

The current density $i_{x}$ as a function of distance x can be calculated as [50,51]:(8)$$i_{x}=I\cdot \left(5.1-5.24\;\mathrm{lg}\left(x\right)\right)$$

This formula is valid for *x* = {1, 8}.

Nickel-coated degreased brass panels (7.5 × 10 cm, Dr.-Ing. Max Schlötter GmbH and Co. KG, Geislingen an der Steige, Germany) were plated under a direct current of 5 A at 55 °C for 7 min with Ir/Ta oxide coated titanium as the anode. To simulate the recycling of the concentrate in the plating process, the buffer mixture and chromium-containing compound were added. The surface tension was adjusted to 52 mN/m by the addition of the commercial surfactant, and the pH was reduced to 3.2 with sulfuric acid (resulting in 8.30 g/L Cr, 89.96 g/L boric acid and 102.68 g/L sulfate). This solution was used in a separate hull cell test.

## 3. Results and Discussion

### 3.1. Electrolyte Recovery with Reverse Osmosis

Although reverse osmosis is known to provide high rejection for heavy metal ions in solution, the complex composition, as well as the targeted high concentrations of the electrolyte to evaluate the recyclability of the RO concentrate, makes this approach challenging. To the knowledge of the authors, no investigations have yet been published on this approach regarding a trivalent chromium electrolyte.

The RO feed water mainly consists of chromium hydroxide sulfate, boric acid, malic acid, sodium saccharin and a succinate surfactant. Concentrations of the main inorganic components and parameters of the RO feed, the final RO concentrate and a typical electrolyte bath are summarized in Table 1. Furthermore, the results of the ICP-OES analysis of all RO concentrates are presented in Figure 3, and a photograph showing the appearance of an RO concentrate and the corresponding permeate is depicted in Figure 4.

Besides chromium, as the main component for the plating process, boric acid was quantified. The wanted high concentration near the solubility limit on the one hand and known relatively low rejection in RO processes on the other hand, made it crucial to monitor the boron concentration. Sulfate was chosen to be the third species to detect, because of its high concentration in the electrolyte and divalent anionic character, contrasting the other two compounds.

Starting with a feed solution of 0.77 g/L Cr, 7.18 g/L boric acid and 7.12 g/L sulfate, a nearly linear concentration increase is observed until cf 4.59 (3.53 g/L Cr, 28.31 g/L sulfate, 26.13 g/L boric acid), when the lower rejection of boric acid becomes apparent. Rejection values are discussed in a subsequent section of this work.

The initial concentration of sulfate was increased by a factor of 9.50, leading to a final concentration of 67.71 g/L, derived from chromium hydroxide sulfate and sulfuric acid. The concentration of 8.40 g/L Cr in the final concentrate (K14.1 with cf = 10.92) exceeds the minimum concentration of 8 g/L Cr needed for the plating electrolyte. Therefore, this concentrate was used for further plating tests. A boric acid concentration of 56.86 g/L was finally observed. This concentration also exceeds the literature values of 36 g/L of boric acid achieved with the same membrane but under basic conditions, and it is in good agreement with the maximum concentration of 62 g/L achieved under acidic conditions of pH 5.2 for pure $B{\left(OH\right)}_{3}$ solutions [52]. It slightly exceeds the reported solubility maximum of 55 g/L at 25 °C for pure boric acids solutions [33] because of the stabilizing effect of present sulfate [41,53]. The concentrates appeared to be clear with a dark blue color (see Figure 4).

### 3.2. Surface Tension

The measurement of the surface tension was used to analyze if the surfactants accumulate in the concentrates. All RO permeates had surface tensions of 72.1 ± 0.1 mN/m, which is in good agreement with the value of pure water (71.9 mN/m) at this temperature. This clearly indicates the absence of significant amounts of surface-active substances (see Figure 5).

On the other hand, RO concentrates showed a systematic decrease in surface tension from 72.0 mN/m to 62.7 mN/m. Surface-active substances, such as the anionic succinate surfactant, are likely to be rejected by the membrane due to their relatively high molecular weight and ionic character. These data indicate the accumulation of the surfactant in the concentrate.

### 3.3. Hull Cell Testing

During the plating, hydrogen gas evolution was observed. The deposited layers appeared crack-free and had a cold chromium hue to them, with a greyish appearance in the areas of high current density (Figure 6, left). An almost clear and even chromium layer in the middle was observed, and no plated layer was present on the right edge of the panel.

The panel plated with the unchanged RO concentrate (panel 1) showed a fine, dark burning line (a, see Figure 6) on the edge of high current density (HCD) next to a brighter field (b) of ca. 3–4 mm width. The first indicates low amounts of conducting salts, whereas the second can be explained by the low concentrations of organic supplements as complexing acid [54]. The greyish tint on the right, next to b, can be caused by the reduction in metal impurities. The width of the plated field of panel 1 was 8.7 cm. According to the standardized procedure, the width of the plated area can only be related to the current density of the plating process for distances from 1 to 8 cm. Here, this translates to current densities of 25.5 $A/{\mathrm{dm}}^{2}$ on the left side of the panel, and to 1.8 $A/{\mathrm{dm}}^{2}$ on its right side (see Equation (8)) [50]. The cold chromium hue was best between 3 and 8 cm, translating to 13.0 $A/{\mathrm{dm}}^{2}$ to 1.8 $A/{\mathrm{dm}}^{2}$, which covers the majority of the working range of 5–15 $A/{\mathrm{dm}}^{2}$ (optimum 10 $A/{\mathrm{dm}}^{2}$) of the initial electrolyte.

The panel was plated with the re-stocked solution (panel 2), the above-described defects a and b disappeared, whereas greyish, vertical lines were visible in the middle area of the panel [55]. The excess of sodium saccharin, which would lead to similar defects, can be excluded as the reason for this observation due to equal concentrations in the plating solutions [51]. For panel 2, the width of the plated field increased from 8.7 to 9.3 cm. The increased buffer concentration hindering hydrogen gas formation and the increased concentration of chelating ligand enabled the chromium deposition even for current densities ≤1.8 $A/{\mathrm{dm}}^{2}$. This is similar to the findings of Zeng et al., who found that chromium can be deposited under lower current densities when the concentration of chelating ligand oxalate is doubled [56]. A chromium deposition for low current densities (LCDs) of ≤3.1 $A/{\mathrm{dm}}^{2}$, as observed here with malic acid as the chelating ligand, is not reported in the literature [56].

### 3.4. Analysis of RO Permeates and Membrane Rejection

Besides the ambition of recovering and reusing valuable chemicals from rinse water, another goal of membrane-based wastewater treatment is the recycling of water supplementing the rinse water tanks. Based on the assumption that the RO permeate would be used to refill rinse water tanks of the same Cr(III) plating line, low concentrations of remaining plating chemicals are tolerable, as they naturally appear in the batch rinses anyway.

The permeates were investigated with ICP-OES, and the remaining chromium, boric acid and sulfate mass concentrations are depicted in Figure 7 in correlation to the cf of the associated concentrate (cf. Figure 3). The concentrations found in the concentrates and permeates were used to calculate the rejection of each species (Equation (7)), depicted in dependency of the actual flux and cf in Figure 7 (right). In this regard, $\beta_{iP}$ results from the magnitude of the diffusion that takes place between the retentate and permeate and from the magnitude of the permeate flux $J_{P}$. By keeping $J_{P}$ constant, it is somehow possible to decouple both effects from one another. Without membrane fouling and eventual changes in membrane surface properties, this approach allows the analysis of diffusion phenomena. As mentioned above, substantial membrane fouling was observed in this work, which might affect solute–membrane interactions during the separation. Furthermore, complex multicomponent diffusion effects have to be considered here (multicomponent diffusivities in mixtures require activity and concentration data).

In Figure 7, several effects occur simultaneously: For low cf values, i.e., 1–3.9, the concentrations of chromium and sulfate found in the permeates increase in an analogous manner. The rejection of chromium started with values of 99.6–99.8%, depending on J, translating into absolute mass concentrations of ≤3.0 mg/L for all fluxes. Sulfate values were 97.1–98.6% (≤0.21 g/L) for cf 1.03, whereas the rejection for boric acid varied between 77.2% and 92.5% (≤2.76 g/L). Here, and in any subsequent experiment, lowering the permeate flux resulted in higher concentrations in the permeate. This flux-dependent increase in rejection is related to the increased permeation of water that is driven by the transmembrane pressure, whereas the permeation of solutes is driven by the concentration difference [57].

For chromium and sulfate, where rejection values were high from the beginning (≥99.8% and ≥97.5% at cf 1.03), permeate concentrations decreased for increasing retentate concentration at a cf of 3.85 to 5.50, translating into concentrations of ≤5.2 mg/L chromium, ≤3.99 g/L boric acid and ≤0.51 g/L sulfate. The comparably high concentration of boric acid is related to a rejection of only 59.5% for $J=10\;L\cdot m^{-2}\cdot h^{-1}$ and 77.2% for $J=30\;L\cdot m^{-2}\cdot h^{-1}$.

From this point on, two phenomena were observed regarding the rejection values. First, the rejection increased in all cases, and second, the rejection became more and more independent of the permeate flux. For example, the rejection of boric acid increased to up to 87.2% for $J=10\;L\cdot m^{-2}\cdot h^{-1}$ and up to 93.8% for $J=30\;L\cdot m^{-2}\cdot h^{-1}$. These values are in agreement with the rejection data reported by Redondo et al. [58]. For chromium and sulfate, the maximum rejection values of 99.9% and 99.6% were observed.

The rejection increase may be related to membrane fouling. The adsorption of organic molecules on the membrane surface may lead to changed surface properties, such as lowered defect density, increasing the rejection on the one hand but decreasing the permeance on the other hand. Results of salt rejection tests and the visual examination of the membrane after the entire experiment are in good agreement with this reasoning (see later section). The commercial electrolyte contains not only succinate-based surfactants but also non-neglectable amounts of malic acid and saccharin, which could cause this behavior. Similar behavior was reported by Cimen during the processing of real acidic chromium wastewater [15].

The increased boron rejection resulted in nearly constant permeate concentrations with increasing concentrations in the retentate. The maximum boric acid concentration was 7.84 g/L, determined at a flux of $3.75\;L\cdot m^{-2}\cdot h^{-1}$ at cf 9.25, which is lower but yet in a similar range to the 8.74 g/L obtained by Bártová et al. for the Filmtec SW30 membrane with a feed stream of primary cooling water from a nuclear power plant [52].

After cf 6.11, when the range of feasible permeate fluxes was reduced, absolute concentrations of all three species in the permeate increased. For the lowest permeate flux of $3.75\;L\cdot m^{-2}\cdot h^{-1}$, values of 5.1 mg/L chromium (cf 10.65), 7.84 g/L boric acid (cf 9.25) and 1.03 g/L sulfate (cf 10.65) were detected. Under economic considerations, such low fluxes are inefficient, but those would only occur in the last phase of a batch treatment. In any case, concentrations in the accumulated permeate would be significantly lower. By integrating the permeate into the refill of the first rinse water tank after the electrolyte bath, the process would benefit from significantly lowered contamination and reduced freshwater consumption.

Because of the complexity of the multicomponent system used in this work (organic and inorganic compounds) and the observed membrane fouling (possible changes in surface properties), we believe that adding a theoretical line to these results is currently impossible and not reasonable. This research highlights the necessity to update current theoretical models in order to improve our understanding of the mechanisms involved in heavy-duty zero liquid discharge applications based on RO membranes.

### 3.5. Analysis of the Process Parameters

The analysis of the RO process parameters and related membrane performance is crucial for the future development of a feasible and applicable treatment. After the criteria of sufficient rejection for all relevant species are met, the maximum possible permeate flux (J) determines the needed membrane area or time to treat a given volume of feed solution. To investigate the process under economically relevant conditions, permeate fluxes of 10 to $30\;L\cdot m^{-2}\cdot h^{-1}$ were realized, and the performance was measured where possible (see Figure 8).

The initial hydraulic pressure at cf 1.03 was 27.1 ± 0.1 bar to achieve a permeate flux of $30\;L\cdot m^{-2}\cdot h^{-1}$ and 13.7 ± 0.0 bar for a flux of $10\;L\cdot m^{-2}\cdot h^{-1}$, respectively. The increase in retentate concentration (cf) resulted in a linear increase in needed hydraulic pressure, finally reaching a pressure of 78.9 ± 0.6 bar for $J=30\;L\cdot m^{-2}\cdot h^{-1}$ and 42.4 ± 0.3 bar for $J=10\;L\cdot m^{-2}\cdot h^{-1}$ at cf 5.50. To further increase the feed concentration without exceeding the maximum admissible hydraulic pressure, permeate fluxes were reduced. The correlation between hydraulic pressure and permeate flux is linear under all studied conditions ($R^{2}\geq 99.7\%$ for cf 1.03 to cf 10.65); the ascending slopes of the hydraulic pressure-flux lines with increasing cf arise from a decrease in permeance (see Equation (4)).

With the obtained hydraulic pressures and osmotic pressures of the RO concentrates, the driving forces were calculated by Δp−Δπ (see Equation (4)). The osmotic pressures of the permeates were neglected due to their comparably small contribution. To determine the influence of concentration polarization and fouling, the membrane’s permeance was determined and is depicted in Figure 9. It shows the cumulative permeance of the two membranes with increasing feed concentration as a function of the calculated driving force.

Taking into account that the influence of the feed’s osmotic pressure is already subtracted, decreasing permeance while the driving force remains constant has to be related to the increasing resistance of the membrane. This can be either membrane fouling, scaling or concentration polarization effects. Taking into consideration that every steady-state experiment was performed consecutively to all previous ones, without membrane cleaning and the presence of organic components in the mixture, fouling is very likely to occur.

Data from experiments at cf 1.03 and cf 2.04 show significant differences in their correlation of permeance with applied driving force from all other (subsequently performed) experiments. Here, the permeance increased with the driving force, although cf 2.04 showed this behavior in attenuated form in comparison to cf 1.03. After the first two series of experiments, permeance decreased with successive increases in cf, from 0.94 ± 0.02$\;L\;\cdot m^{-2}\cdot h^{-1}\cdot {\mathrm{bar}}^{-1}$ (cf 3.08) to 0.25 ± 0.02 $L\;\cdot m^{-2}\cdot h^{-1}\cdot {\mathrm{bar}}^{-1}$ (cf 10.92) and was nearly independent of the driving force. This indicates that concentration polarization has only a minor influence, and fouling is the dominant reason for decreasing permeance.

Concluding the made observations, the recovery of chromium was accompanied by the decrease of permeance from 0.99 ± 0.15 $L\;\cdot m^{-2}\cdot h^{-1}\cdot {\mathrm{bar}}^{-1}$ down to 0.25 ± 0.2 $L\;\cdot m^{-2}\cdot h^{-1}\cdot {\mathrm{bar}}^{-1}$. However, because a large part of the overall experiment had been performed not in concentration mode but in steady-state mode, the challenge of the membrane with fouling substances was much higher than it would be in an actual batch mode treatment in concentration mode. Specifically, circulating feed and permeate overnight with reduced flux must have promoted fouling.

### 3.6. Salt Rejection Test and Membrane Observation

The results of the salt rejection tests are summarized in Table 3. It shows that membrane permeance was more than halved. To achieve a flux of 30.0 $L\cdot m^{-2}\cdot h^{-1}$, the required hydraulic pressure increased from 34.2 to 55.4 bar. Applying a similar hydraulic pressure as before the RO experiment led to half of the flux.

The optical examination of the membrane (see Figure 10) showed severe forms of fouling. A dense black fouling layer was observed, which appeared gel-like when wet. However, we cannot exclude an additional membrane compaction effect. A differentiation between membrane fouling and membrane compaction cannot be clearly derived from our data.

A detailed analysis of the layer is part of future investigations, with an aim to reveal whether the permeance decrease is mainly due to fouling, scaling or a superposition of both. The integration of a suited cleaning procedure for the planned process must be investigated as well.

## 4. Conclusions

With regard to the European Green Deal as the main roadmap of the EU Commission for sustainability transformation, the accumulation of rinse water from plating industries is cost-intensive and not environmentally acceptable. This study shows that a Cr(III) electrolyte containing a variety of inorganic (sulfate, chromium, boric acid) and organic (malic acid, saccharin, succinic surfactant) components can be successfully recovered from simulated electroplating rinse water in a single-step batch RO process and directly reused for electroplating purposes. In this context, Hull cell tests verified the ability to deposit a cold-hued chromium layer from the RO product with and without further addition of plating components in a wide current density range, including the optimum conditions for the electrolyte. This proof of reusability of the RO concentrate has not been reported for trivalent chromium electrolytes in the literature yet. Although the presented experimental RO procedure promoted membrane fouling, due to long periods of steady-state conditions and low permeate flux, excellent rejection for sulfate, chromium and boric acid was observed. The close observation of process parameters suggested that the treatment process itself is strongly dependent on the osmotic pressure of the retentate, and the role of concentration polarization could not be clearly identified. While only minor chromium traces could be measured in permeates, these fractions still contain some amounts of sulfate and boric acid. According to BIA’s development team, a combined permeate, as it would occur in a batch treatment, can be seriously considered for reuse within the plating process. The progess achieved in this work should be regarded as a promising step toward boosting circularity in industrial electroplating operations.

Future research must examine the nature of the observed layer on the membrane surface. The complex composition of the electrolyte neither excludes fouling nor scaling or a mixed blocking layer. After the characterization of the layer, an effective cleaning procedure has to be developed to recover membrane permeance. To achieve the goal of the IntelWATT project, the applicability at a larger scale has to be determined.

## Figures and Tables

**Figure 1 membranes-12-00853-f001:**
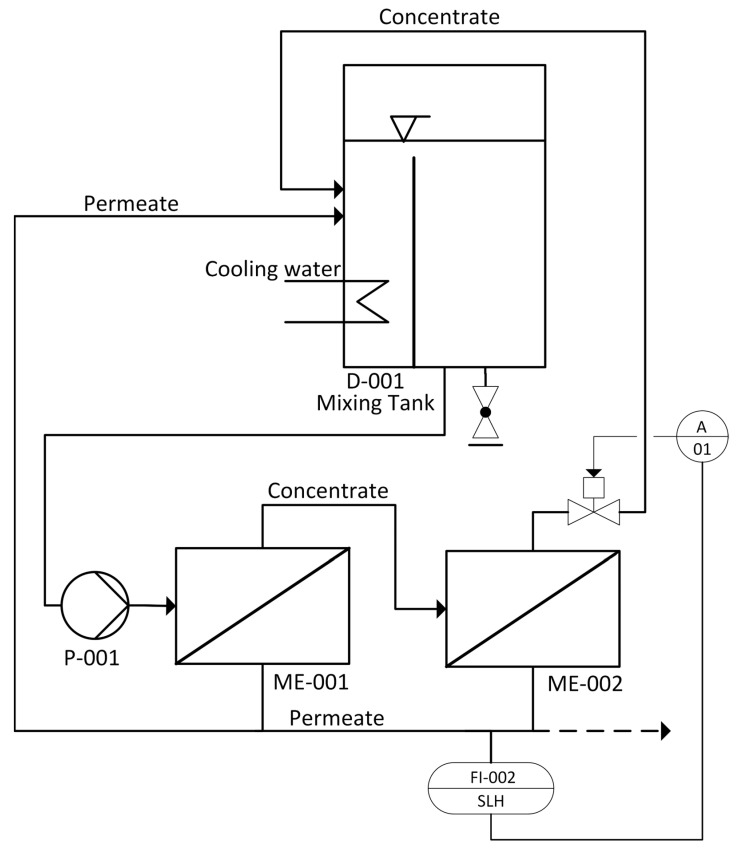
Process flow diagram of the ROLU used for reverse osmosis experiments on a lab scale. The concentration mode uses the dashed pipeline.

**Figure 2 membranes-12-00853-f002:**
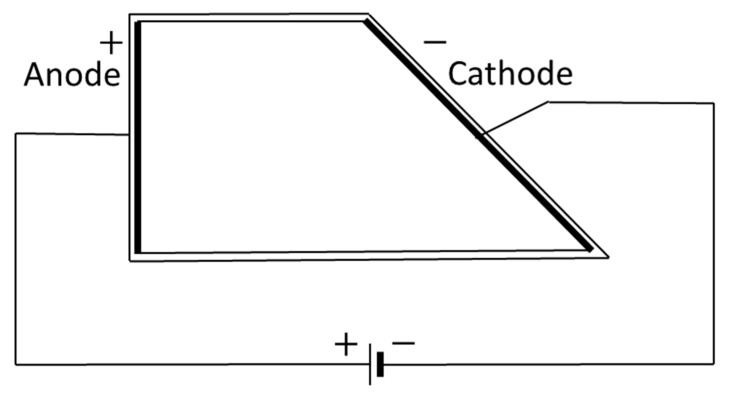
Electric circuit of a rectangular Hull cell. The anode and cathode panels are depicted in thicker lines.

**Figure 3 membranes-12-00853-f003:**
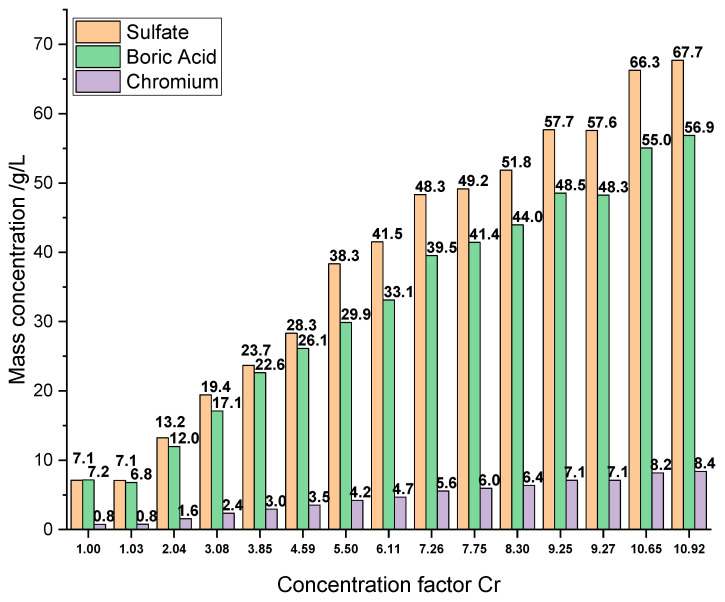
Mass concentration of sulfate, boric acid and chromium in the RO feed (cf 1.00) and concentrates K1.1–K14.1 (cf 1.03–10.92) as a function of the concentration factor (cf) of chromium in each stage of the series of consecutive experiments.

**Figure 4 membranes-12-00853-f004:**
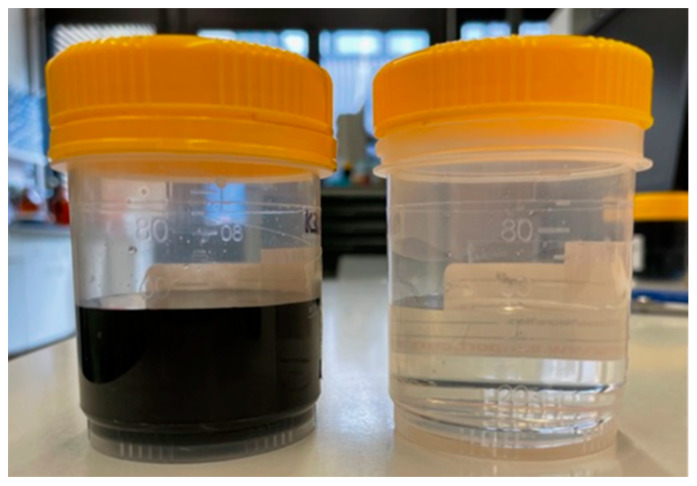
Photo of RO concentrate K3.1 (left) and permeate P3.1 (right). The concentrates show a dark blue color.

**Figure 5 membranes-12-00853-f005:**
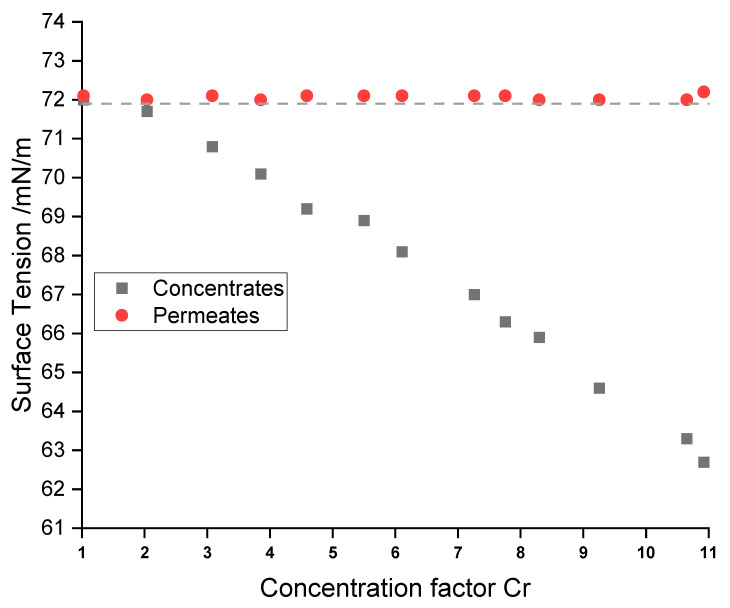
Surface tension of the RO permeates and concentrates with increasing concentration factor (cf) of chromium. The dashed line marks the surface tension of pure water at 25 °C.

**Figure 6 membranes-12-00853-f006:**
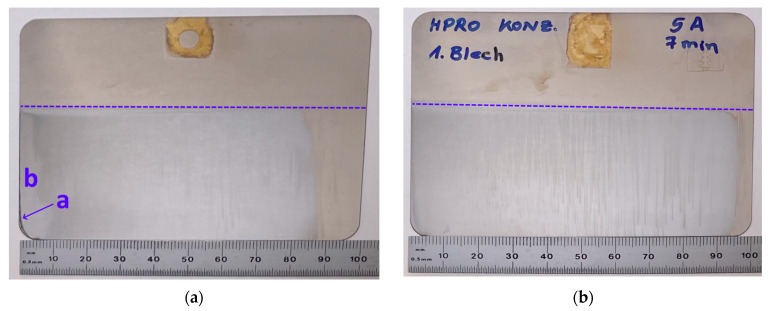
Hull cell test panels after coating: (**a**) Panel coated with RO concentrate without and (**b**) with further addition of plating agents. The dashed line indicates the submerged area of the panel.

**Figure 7 membranes-12-00853-f007:**
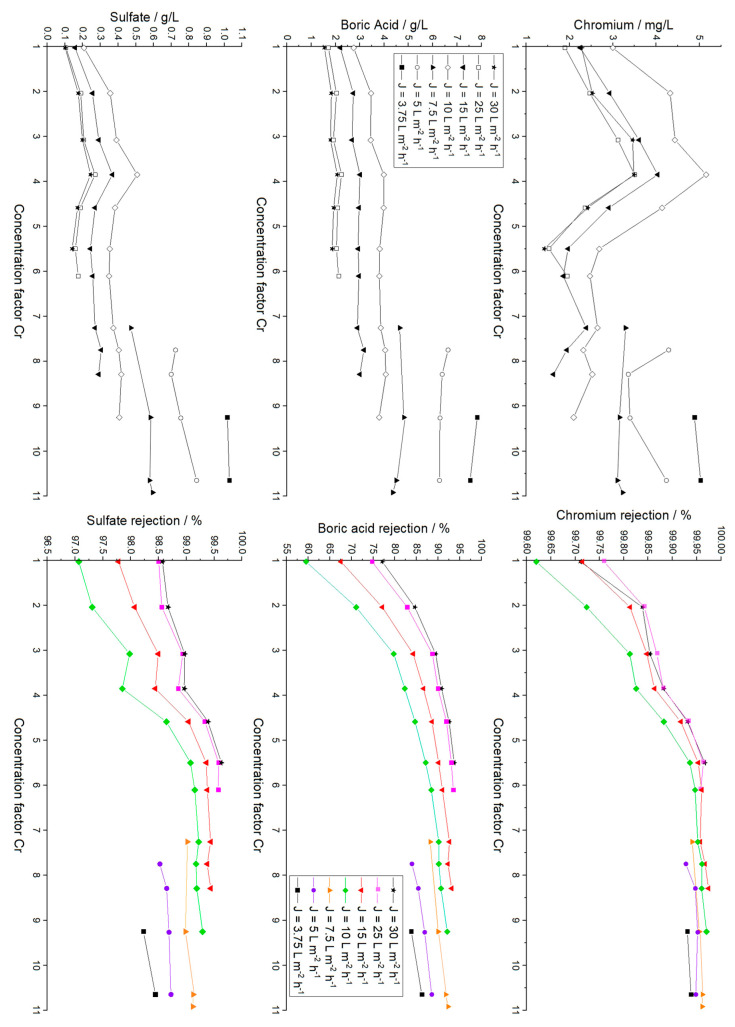
Mass concentration of main inorganic components in RO permeates for different permeate fluxes J and chromium concentrations factors (left) and corresponding solute rejection values (right). The errors for the mass concentrations in permeates are as follows: Cr: ≤2.3%, sulfate: ≤3.6%, boric acid: ≤4.6%. The errors for the rejections are as follows: Cr: ≤3.5%, sulfate: ≤4.3%, boric acid: ≤4.0%.

**Figure 8 membranes-12-00853-f008:**
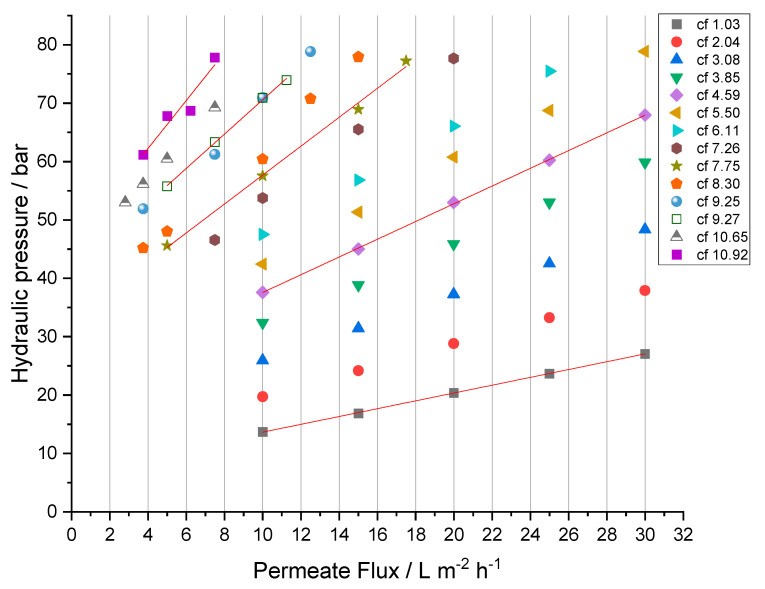
Correlation between hydraulic pressure and permeate flux for the different concentration factors of chromium (cf). Linear regression lines are added for visual guidance.

**Figure 9 membranes-12-00853-f009:**
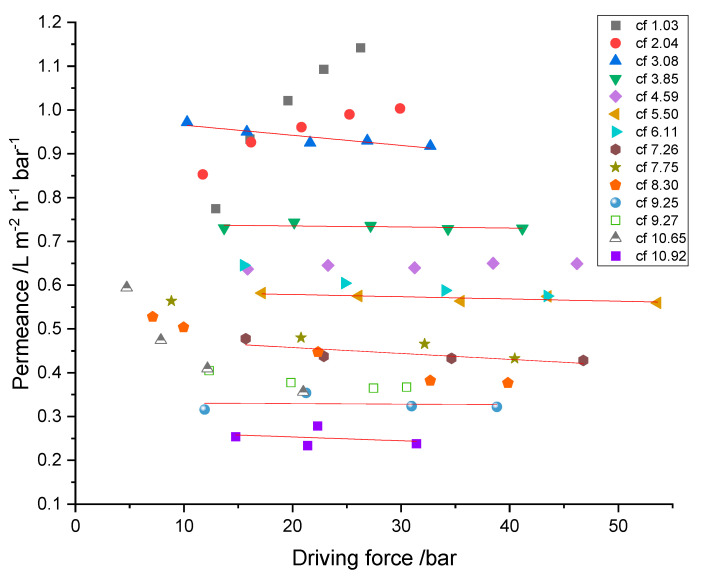
Membrane permeance at different feed concentrations corresponding to different Cr concentration factors (cf) in relation to the driving force. Linear regression lines are shown for visual guidance.

**Figure 10 membranes-12-00853-f010:**
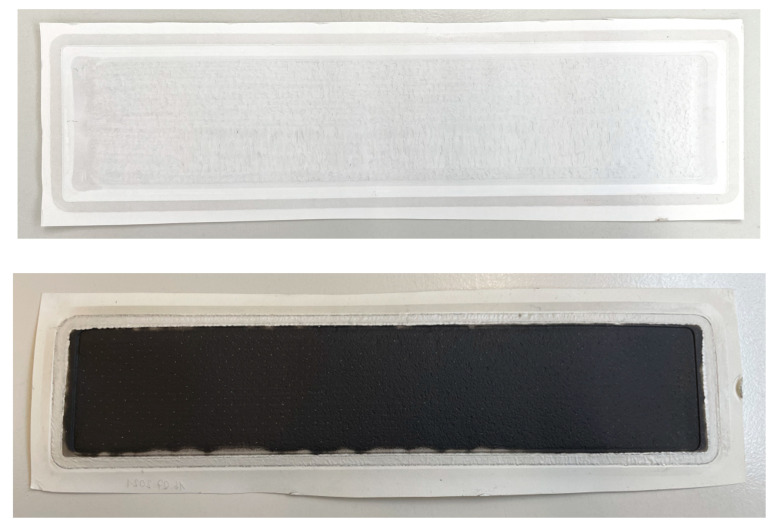
RO membrane before (**above**) and after (**below**) RO experiment. The black fouling layer was observed after 60 days of use.

**Table 1 membranes-12-00853-t001:** Concentrations of main inorganic components, surface tension and pH value of the Cr(III) electrolyte, RO feed, as well as the final RO concentrate and concentration factors of the monitored substances in that concentrate.

	Β(Cr)/g/L	$\beta (B{\left(OH\right)}_{3})/g/L$	$\beta (SO_{4}{}^{2-})/g/L$	γ/mN/m	pH
Cr(III) electrolyte *	8–12	80–110	80–150	30–50	3.5–3.9
RO feed	0.77	7.18	7.12	72.0	4.7
Final concentrate K14.1	8.40	56.87	67.71	63.7	4.02
Concentration factor	10.9	7.92	9.51	-	-

* information provided by BIA Kunststoff- und Galvanotechnik GmbH & Co. KG.

**Table 2 membranes-12-00853-t002:** Membrane specifications and experimental conditions of the RO experiment.

**Membrane**	Filmtec SW30-2540, DuPont	
**Membrane type**	Polyamide thin-film composite	
**Membrane area**	Two sheets, each 80 cm^2^	
	From data sheet	Observed in RO experiment
**NaCl rejection**	99.4%	93.0%
**Maximum pressure**	69 bar	80 bar
**Temperature**	25 °C	25 °C
**pH range**	2–11	3.8–4.7

**Table 3 membranes-12-00853-t003:** Salt rejection, permeate flux, hydraulic pressure and membrane permeance before and after the RO experiment.

	$R_{NaCl\;}/\%$	$J/L\cdot m^{-2}\cdot h^{-1}$	$p_{hydraulic\;}/\mathrm{bar}$	$A/L\;\cdot m^{-2}\cdot h^{-1}\cdot bar^{-1}$
Pre RO	93.0	30.0	34.2	1.987
Post RO $\left(J_{pre}=J_{post}\right)$	98.3	30.0	55.4	0.826
Post RO $\left(p_{pre}=p_{post}\right)$	97.8	15.0	35.0	0.943

## Data Availability

Not applicable.

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
