# Peer review of "A Reverse Osmosis Process to Recover and Recycle Trivalent Chromium from Electroplating Wastewater"

_membranes, 2022, doi:10.3390/membranes12090853_

Round 1
Reviewer 1 Report
· Abstract: well written, concise and provide complete information for bird's eye view reader about the content of the paper.
· Methodology: more explanation on how the artificial wastewater was prepared is needed
· General Comments:
o Kindly use mathematical symbols as multiplication signs (×) or (.) for the units instead of (*)
o Figure 1. Change Piping and instrumentation diagram to process flow diagram
o I suggest adding a list of abbreviations to the paper to ease following the symbols
o In figure 10 it would have been better for the sake of comparison to include the picture of the membrane before the fouling next to the fouled one to show the severity of the dense layer formed
Author Response
Dear reviewer,
thank you very much for your helpful suggesttions.
We changed the manuscript accordingly.
We hope these changes meet your expectations.
Sincerely,
Roxanne Engstler and co-authors.

Reviewer 2 Report
A paper entitled “A straightforward approach for recovery and recycling of trivalent chromium from electroplating wastewater.” demonstrated reverse osmosis (RO) experiment to recycle the metal ion from electroplating wastewater. In addition, the authors also demonstrated the electroplating process using the recycled solution after RO process, and obtained a good data.
However, because of poor discussion without theoretical discussion and effective graph, the authors are strongly recommended to dramatically revise the manuscript for publication of this paper to Membranes journal which recently has a high impact factor about 4.5. The following comments should be concerned to improve paper quality at least.
(C1) Figure 9:
Figure 9 becomes Figure 2. Please revise the mistake.
(C2) Figure 9:
To obtain the permeance, did you consider the outlet concentration? When increasing cf, the effect of outlet concentration would be increased. Generally, average mean or logarithmic mean will be used to calculate the permeance. Please considered.
(C3) Supplemental data:
Please add the supplemental result data for all experiment. The data will be useful for the readers for further analyzing.
(C4) Figure:
Please write all figures using x-axis of normal magnification using general number such as 1 2 3 4 like Figure 8. (Please do not list the concentration factor for x-axis such as 1.03, 2.04 ,3.08 and so on)
(C5) Figure 8:
Please change the x-axis and y-axis. In addition, the minimum value of hydraulic pressure should be zero. In addition, the cross section between y-axis (hydraulic pressure) and the respective line of the data must be the osmotic pressure of the solution with considering cf inside the lab test unit. In this case, the permeance in figure 9 would be changed when using the osmotic pressures from the cross sectional values above. Please check which is preferable.
(C6) Concentration factor:
In Figure 3, the cf should be 10.5 (= 8.4/0.77) instead of 10.92. please check the calculation again. In addition, please add an equation to explain concentration factor (cf) clearly.
(C7) Table 3:
This result is interesting. However, is it possible to distinguish the difference between fouling and compaction. which is main reason for this result? please explain and discuss in the manuscript.
(C8) Performance analysis:
In all figure, there is no theoretical analysis, that make the discussion poor. For example, in Figure 7, rejection increased with increasing Jw with same cf. in this case, these data are theoretically acceptable? If possible, please add the theoretical line in the figure using solution diffusion model or spiegler kedem model and so on.
(C9) Performance analysis:
In Figure 8, the RO permeate concentration increased with increasing cf. However, rejection increased with increasing cf. The rejection = (1-(Cp/Cf))×100 where the concentrations are inlet ones as shown in Eq. (6). In this case, feed concentration was changed? Please explain.
(C10) Jw control during the experiment:
Please add the reason why the same Jw experiment is useful to discuss the rejection and so on in your experiment for the readers.
(C11) Table 3:
Table 1 in page 15 should be Table 3. please revise the mistake.
Author Response
Dear reviewer,
thank you for your helpful suggestions. We changed the manuscript accordingly.
We addressed your suggestions point-by-point in the attached letter.
Sincerely,
Roxanne Engstler and co-authors.

Round 2
Reviewer 2 Report
Dear Authors,
Firstly, please show where the authors revised the manuscript with their revision history, because the reviewers can not understand where the authors revised.
Best regards,